# Craniocervical Morphometry in Pomeranians—Part I: Intra-Observer, Interobserver, and Intermodality (CT vs. MRI) Agreement

**DOI:** 10.3390/ani14131854

**Published:** 2024-06-22

**Authors:** Koen Santifort, Sophie Bellekom, Ines Carrera, Paul Mandigers

**Affiliations:** 1IVC Evidensia Referral Hospital Arnhem, 6825 MB Arnhem, The Netherlands; 2IVC Evidensia Referral Hospital Hart van Brabant, 5144 AM Waalwijk, The Netherlands; 3Expertise Centre of Genetics, Department of Clinical Sciences, Faculty of Veterinary Medicine, Utrecht University, 3584 CL Utrecht, The Netherlands; 4Vet Oracle Teleradiology, Norfolk IP22 4ER, UK

**Keywords:** interobserver reliability, intra-observer agreement, caudal cranial fossa, magnetic resonance imaging, computed tomography

## Abstract

**Simple Summary:**

Numerous diagnostic imaging-based studies (computed tomography (CT) and magnetic resonance imaging (MRI)) have focused on the shapes and sizes of the skull and neck of small and toy breed dogs in relation to disorders of the skull, vertebral column, and spinal cord. No such studies have particularly focused on the Pomeranian dog breed. The aims of Part I of this two-part study were to assess the intra-observer, interobserver and intermodality (CT vs. MRI) agreement for several features of the skull and vertebrae on CT and MRI studies. Multiple observers performed classifications of CM/SM and measurements on CT and MRI images of 99 Pomeranians. The results indicated that the reliability of classification of CM/SM between observers varied between different levels of experience and that the interobserver reliability of various measurements was imperfect. These results must be taken into account when assessing the results of future studies and when clinical cases are classified based on grading schemes.

**Abstract:**

Background: Currently, there are no diagnostic imaging-based studies that have focused specifically on the craniocervical morphology of Pomeranian dogs in relation to Chiari-like malformation and syringomyelia (CM/SM). The aims of Part I of this two-part study are to assess the intra-observer, interobserver, and intermodality reliability and agreement for various aspects of the craniocervical morphology of Pomeranians with and without CM/SM. Methods: Prospectively, Pomeranians were included that underwent both CT and MRI studies of the head and cervicothoracic vertebral column. Two observers (experienced and novice) independently performed CM and SM classifications, qualitative assessments, and quantitative measurements. The experienced observer performed these assessments twice. A third observer (experienced) performed CM and SM normal or abnormal classifications. Results: Ninety-nine (99) dogs were included. Interobserver reliability was influenced by observer experience level. For the experienced versus novice observers, substantial interobserver agreement was found for classification of SM as normal or abnormal (Cohen’s kappa = 0.63), while interobserver agreement was fair for classification of SM as normal or abnormal (Cohen’s kappa = 0.31). Interobserver, intra-observer, and intermodality reliability were variable for different measurements and assessments, and best overall for the CT-based measurements. Conclusions: Interobserver reliability and agreement results should be taken into account in the evaluation of results of future studies as well as the evaluation of imaging studies of Pomeranians presented clinically.

## 1. Introduction

Chiari-like malformation (CM) and syringomyelia (SM) are disorders that have been documented in various dog breeds, including the Cavalier King Charles Spaniel (CKCS), Griffon Bruxellois, and other small and toy breed dogs [1,2,3,4,5]. In the CKCS, many imaging features, mostly based on magnetic resonance imaging (MRI), have been linked to CM/SM and have provided insights into pathophysiology as well as clinically relevant characteristics that can be communicated to owners and breeders. A recent review covers a variety of imaging features that have been studied in relationship to CM/SM [6].

A fairly large number of diagnostic imaging-based (computed tomography (CT) and MRI) studies have focused on the morphology of the skull and craniocervical region of the CKCS and other mostly brachycephalic and small or toy breed dogs in relation to disorders such as atlantoaxial instability, atlantoaxial overlapping, and CM/SM [1,2,3,4,5,6,7,8,9,10,11,12,13,14,15,16,17]. These studies have provided valuable information on the pathogenesis of and relationships between these disorders.

Recently, we reported the prevalence of CM/SM, owner-reported clinical signs and the associations thereof with a CM/SM grading scheme in Pomeranians [18]. However, morphological studies that specifically focus on the Pomeranian dog breed that evaluate the relationship of anatomical (imaging-based) features with CM/SM status are lacking. The aims of Part I of this two-part study were to assess the intra-observer, interobserver, and intermodality reliability and agreement for various aspects of the skull and craniocervical region of Pomeranians with and without CM/SM, by means of CT and MRI.

## 2. Materials and Methods

For this prospective study, Pomeranians were included that underwent both CT and MRI studies of the head and cervicothoracic vertebral column at two institutions (IVC Evidensia Small Animal Hospital Arnhem and IVC Evidensia Small Animal Hospital Hart van Brabant) during the period of February 2022 to October 2023. All owners agreed to participate in this study and the informed consent of the owner was obtained. The study was approved by the Animal Welfare Body Utrecht, Utrecht University, The Netherlands. The studies were conducted in accordance with the local legislation and institutional requirements.

Exclusion criteria included dogs with a prior history of, or that were diagnosed with central nervous system (CNS) disease on MRI other than one or more of the following (i.e., dogs affected by the following disorders were included): CM/SM, ventriculomegaly, supracollicular fluid accumulation, findings related to craniocervical junction abnormalities (CJA) (e.g., atlanto-occipital overlapping (AOO), dorsal constriction at C1/C2 (atlantoaxial band (AAB)), atlantoaxial instability (AAI, also referred to as atlantoaxial subluxation), or non-structural disorders such as epilepsy or paroxysmal dyskinesia. Dogs with MRI or CT studies with artifacts or insufficient image quality that did not allow for accurate assessments or measurements were also excluded.

MRI and CT studies were performed under general anesthesia (individualized anesthetic protocols) with a high-field MRI scanner (1.5T Canon Vantage Elan, The Netherlands) and a 16-slice CT scanner (Siemens SOMATOM.go, The Netherlands). Dogs were positioned in sternal recumbency on the horizontal surface of the table with the head in a flexible coil (MRI) or a head rest (CT), both resulting in elevation of the head by about 2–3 cm above the table (Figure 1). MRI sequences obtained included sagittal T2W (echo time (TE) 110 ms, repetition time (TR) 2.6 s, 2.5 mm slice thickness, 256 × 320 matrix), sagittal T1W (TE 10 ms, TR 0.5 s, 2.5 mm slices, 256 × 320 matrix), transverse T2W of the brain (TE 115 ms, TR 4.1 s, 3.0 mm slices, 160 × 192 matrix), transverse T2W of the cervical spinal cord (TE 115 ms, TR 4.1 s, 3.0 mm slices, 160 × 192 matrix) and transverse T1W of the cervical spinal cord (TE 10 ms, TR 0.4 s, 3.0 mm slices, 160 × 192 matrix). Transverse slices at the level of the cervical spinal cord were adjusted to center the syrinx, if visible. In dogs without a visible syrinx on sagittal images, transverse images were acquired at the level of the C2-C3 vertebrae. CT scans were performed with the following parameters: 130 kVp tube voltage, 80 and 220 mAs tube current, 256 × 256 image matrix, 0.6 and 0.8 mm slice thickness, 0.4 and 0.6 mm slice increment, 1.0 s rotation time, and a pitch of 0.6. A bone algorithm was used for image reconstruction in transverse, dorsal, and sagittal planes. Three-dimensional reconstructions including soft tissues were obtained by use of imaging software (RadiAnt DICOM Viewer [Software version 2023.1]).

### 2.1. CT and MRI Evaluation

Two observers (KS (Diplomate European College of Veterinary Neurology) and SB (resident European College of Veterinary Diagnostic Imaging)) independently reviewed the MRI and CT imaging studies and performed CM and SM classifications, qualitative assessments and quantitative measurements using imaging software (RadiAnt DICOM Viewer [Software version 2023.1]). One observer was experienced in classification of CM and the classification and localization of SM (KS); the other was a novice (SB). A third observer (PM (Diplomate European College of Veterinary Neurology; experienced) performed CM and SM normal or abnormal classifications only to assess experienced interobserver reliability. For assessments and measurements other than CM and SM classification, the observers (KS and SB) did not have specific prior experience as the assessments were tailored to this study. Instructions (see Section 2.1.1, Section 2.1.2, Section 2.1.3 and Section 2.1.4) and example images (Figure 2 and Figure 3) were available for the observers prior to the assessments. The experienced observer (KS) performed all these assessments and measurements twice. All sequences and reconstructions were available to the observers for evaluation.

#### 2.1.1. Classification of Chiari-like Malformation (CM)

Images were evaluated to assess the presence or absence of CM by evaluating the shape of the cerebellum and position of the caudoventral cerebellum (uvula). CM was classified as described previously [18], where CM normal = CM0 and CM abnormal = CM1 and CM2.

The line of the foramen magnum was defined as a straight line between the most ventral aspect of the supraoccipital bone and the most caudal aspect of the basioccipital bone on sagittal MR images (Figure 2).

#### 2.1.2. Classification of Syringomyelia (SM)

Images were evaluated to assess the presence or absence of SM in the spinal cord. SM was defined as a well-demarcated intramedullary lesion (or lesions) associated with the central canal of the spinal cord, hyperintense on T2W and hypointense on T1W images. SM was classified as previously described [18], where SM normal = SM0 and SM abnormal = SM1 and SM2.

Additionally, the syrinx location (when present) was noted as follows: cervical, thoracic, extensive (both cervical and thoracic, continuous), or multifocal (both cervical and thoracic, discontinuous).

#### 2.1.3. MRI-Based Qualitative Parameters

The following additional qualitative parameters were assessed on midsagittal T2-weighted MR images:Presence of cerebrospinal fluid (CSF) signal between the cerebellum and the supraoccipitum (yes/no);Presence of CSF signal at the ventral aspect of the cervicomedullary junction (yes/no);Presence of CSF signal at the dorsal aspect of the cervicomedullary junction (yes/no).

#### 2.1.4. Quantitative MRI- and CT-Based Measurements

Skull and vertebral quantitative morphometric measurements, performed on midsagittal CT reconstructions and midsagittal MR images, included the following.

Cranial fossa (Figure 2):Distance between the os tentorium cerebelli and the dorsum sellae (yellow line);Length of the clivus (dorsum sella turcica to ventral margin of foramen magnum) (green line);Height of the foramen magnum (‘foramen magnum line’, red line);Distance between the cranial tip of dorsal arch of the atlas and the foramen magnum line (blue line)—when negative, these dogs were interpreted to have atlanto-occipital overlapping;Area between the yellow line and the red line and osseous structures (caudal cranial fossa area, Area 1);Area rostral to the yellow line (rostral and middle cranial fossa area, Area 2).

Craniocervical junction (Figure 3):Angle between the line from the caudal tip of the basioccipital bone to the cranial tip of the dens axis (red line) and the line from the ventral aspect of the supraoccipital bone to the cranial tip of the dens axis (green line) (Angle 1);Angle between the line from the caudal tip of the basioccipital bone to the cranial tip of the dens axis (red line) and the line from the caudal tip of the basioccipital bone to the midpoint of the caudal endplate of the axis (blue line) (Angle 2).

**Figure 2 animals-14-01854-f002:**
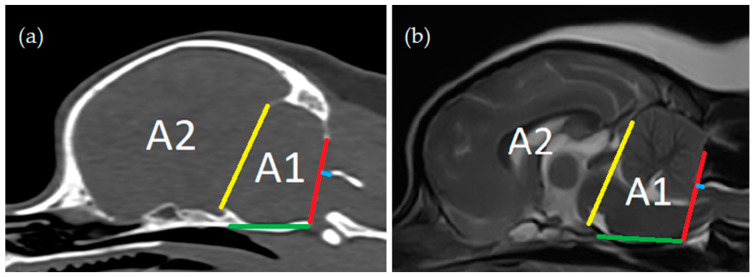
Cranial fossa measurements: (**a**) computed tomography, midsagittal plane reconstruction (bone window); (**b**) magnetic resonance imaging—T2-weighted midsagittal plane. Distance between the os tentorium cerebelli and the dorsum sellae (yellow line); length of the clivus (green line); height of the foramen magnum (‘foramen magnum line’, red line); distance between cranial tip of dorsal arch of the atlas and the foramen magnum line (blue line); area between the yellow line and the red line and osseous structures (caudal cranial fossa area, Area 1); area rostral to the yellow line (rostral and middle cranial fossa area, Area 2).

**Figure 3 animals-14-01854-f003:**
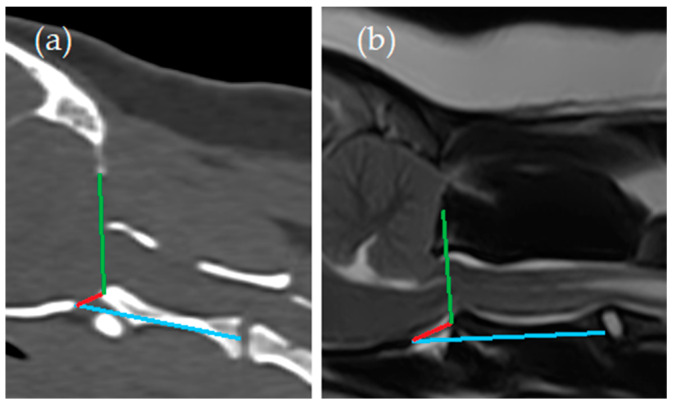
Craniocervical junction measurements: (**a**) computed tomography, midsagittal plane reconstruction (bone window); (**b**) magnetic resonance imaging—T2-weighted midsagittal plane. Angle between the line from the caudal tip of the basioccipital bone to the cranial tip of the dens axis (red line) and the line from the ventral aspect of the supraoccipital bone to the cranial tip of the dens axis (green line) (Angle 1); angle between the line from the caudal tip of the basioccipital bone to the cranial tip of the dens axis (red line) and the line from the caudal tip of the basioccipital bone to the midpoint of the caudal endplate of the axis (blue line) (Angle 2).

### 2.2. Statistical Analysis

Descriptive statistics are reported. Data were tested for normality using a Kolmogorov–Smirnov test. Intra- and interobserver reliability was assessed for the classifications (nominal data: Cohen’s kappa, ordinal data: weighted kappa), additional qualitative assessments (nominal data: Cohen’s kappa), and quantitative measurements (Intraclass Correlation Coefficient (ICC)—interobserver agreement: model = two-way random effects, type = agreement, unit = single rating; intra-observer agreement: model = two-way mixed-effects, type = agreement, unit = average). Agreement between MRI- and CT-based quantitative measurements was assessed with ICC (model = two-way random effects, type = agreement, unit = single rating). Statistical analyses were performed using Microsoft Excel^®^ v2404 and R v4.3.1.

#### 2.2.1. Interpretation of Cohen’s Kappa and Weighted Kappa [19]

Kappa 0.00–0.20 = poor—slight agreement;Kappa 0.21–0.40 = fair agreement;Kappa 0.41–0.60 = moderate agreement;Kappa 0.61–0.80 = substantial agreement;Kappa 0.81–1.00 = almost perfect agreement.

#### 2.2.2. Interpretation of Intraclass Correlation Coefficient (ICC) [20]

Values < 0.5 = poor reliability;Values 0.5–0.75 = moderate reliability;Values 0.75–0.9 = good reliability;Values > 0.90 = excellent reliability.

## 3. Results

### 3.1. Study Population

A total of 112 dogs were eligible for inclusion in the study, of which 13 were subsequently excluded due artifacts on imaging studies, insufficient image quality, and diagnosis of intracranial pathology (e.g., meningoencephalitis of unknown origin). Ninety-nine (99) dogs were, therefore, included in the study. Table 1 includes the characteristics of the study population.

### 3.2. Interobserver Reliability and Agreement Percentages

#### 3.2.1. Interobserver Reliability and Agreement Percentages for Experienced versus Novice Observers

The interobserver reliability and agreement (experienced versus novice) for the classification of CM and SM, MRI-based qualitative parameters, and quantitative MRI- and CT-based measurements as determined by Cohen’s kappa, ICC, and percentage agreement are included in Table 2 and Table 3, respectively.

Substantial interobserver agreement was found for classification of SM as normal or abnormal, presence of CSF signal at the ventral aspect of the cervicomedullary junction, and syrinx location. All other assessments showed slight (presence of CSF signal between the cerebellum and the supraoccipitum, presence of CSF signal at the dorsal aspect of the cervicomedullary junction), fair (CM classification, CM classification as normal or abnormal), or moderate (SM classification) interobserver agreement.

For the MRI-based measurements, good interobserver reliability was found for measurement of Area 1 (caudal cranial fossa), Area 2 (middle and rostral cranial fossa), and Angle 2. All other measurements showed a poor (height of the foramen magnum and Angle 1), or moderate (distance between the os tentorium cerebelli and the dorsum sellae, length of the clivus, and distance between cranial tip of dorsal arch of the atlas and the foramen magnum line) interobserver reliability.

For the CT-based measurements, excellent interobserver reliability was found for measurement of the height of the foramen magnum, distance between cranial tip of dorsal arch of the atlas and the foramen magnum line, and Area 2 (rostral and middle cranial fossa). All other measurements showed a good interobserver reliability (distance between the os tentorium cerebelli and the dorsum sellae, length of the clivus, Area 1 (caudal cranial fossa), Angle 1, and Angle 2).

#### 3.2.2. Interobserver Reliability and Agreement Percentages for Experienced versus Experienced Observers for CM and SM Classification

The interobserver reliability and agreement (experienced versus experienced) for the classification of CM and SM as normal or abnormal as determined by Cohen’s kappa and percentage agreement are included in Table 4.

Almost perfect interobserver agreement was found for classification of SM as normal versus abnormal, and substantial agreement was found for classification of CM as normal or abnormal.

### 3.3. Intra-Observer Reliability and Agreement Percentages

The intra-observer reliability and agreement (experienced) for the classification of CM and SM, MRI-based qualitative parameters, and quantitative MRI- and CT-based measurements as determined by Cohen’s kappa, ICC, and percentage agreement are included in Table 5 and Table 6, respectively.

Intra-observer agreement was almost perfect for most assessments (CM classification as normal or abnormal, SM classification, SM classification as normal or abnormal, presence of CSF signal at the dorsal aspect of the cervicomedullary junction, and syrinx location), and substantial for CM classification, presence of CSF signal between the cerebellum and the supraoccipitum, and presence of CSF signal at the ventral aspect of the cervicomedullary junction.

For the MRI-based measurements, excellent intra-observer reliability was found for measurement of the height of the foramen magnum, distance between cranial tip of dorsal arch of the atlas and the foramen magnum line, Area 2 (rostral and middle cranial fossa), and Angle 2.

All other measurements showed a good intra-observer reliability (distance between the os tentorium cerebelli and the dorsum sellae, length of the clivus, Area 1 (caudal cranial fossa), and Angle 1.

For the CT-based measurements, excellent intra-observer reliability was found for all measurements.

### 3.4. Agreement between MRI- and CT-Based Quantitative Measurements

The intermodality reliability for quantitative MRI- versus CT-based measurements as determined by ICC are included in Table 7.

Moderate intermodality reliability was found for measurement of the length of the clivus, height of the foramen magnum, and Area 2 (middle and rostral cranial fossa). All other measurements showed a poor intermodality reliability (distance between the os tentorium cerebelli and the dorsum sellae, distance between cranial tip of dorsal arch of the atlas and the foramen magnum line, Area 1 (caudal cranial fossa), Angle 1, and Angle 2).

## 4. Discussion

This two-part study is the first large study specifically focusing on the Pomeranian dog breed that evaluates the intra-observer, interobserver and intermodality reliability and agreement for various aspects of the skull and craniocervical region of Pomeranians with and without CM/SM, by means of CT and MRI. The following sections discuss the results of this study in the light of previously published studies.

### 4.1. CM and SM Classification

Interobserver reliability for CM classification was only fair (with a total agreement percentage for classification as CM normal versus abnormal of 70%) for the novice versus experienced observers. The reliability improved markedly to substantially for the experienced versus another experienced observer (total agreement percentage of 91%). These results are similar to those of a previous study that assessed the interobserver agreement for grading CM in the CKCS breed amongst different categories of observers (Diplomates American College of Veterinary Radiology (ACVR), second-year residents ACVR, and interns) [21]. Based on our results as well as those of Weber et al. [21], it can be concluded that the reliability of CM classification is not flawless.

This is not surprising as the classification schemes used to grade CM are, to a degree, subjective in nature. Indeed, the definition of CM is rather vague and variable between publications, with a recent review stating that CM is ‘a complex malformation that results in caudal fossa crowding and displacement of the cerebellum into the foramen magnum’ [22]. The many imaging features related to CM reflect this complexity [6]. Weber et al. found that differentiating between CM0 and CM1 was less consistent than identifying cerebellar herniation (CM2), which was consistently identified by experienced observers on both MRI and CT images [21,23].

On the other hand, interobserver reliability for SM classification as normal versus abnormal was substantial (experienced versus novice observers) to almost perfect (experienced versus experienced observers). This classification (i.e., normal versus abnormal) is promising for two reasons. One is that Weber et al. reported low Cohen’s kappa values for the SM grading scheme they employed to differentiate SM1 from SM2, while concluding on the presence or absence of SM (SM0 versus SM1 or SM2) was more consistent [16]. Second, recent work in the CKCS breed indicates that selection for breeding of SM normal rather than SM abnormal dogs would likely contribute more to reducing the prevalence of SM in the CKCSs better than would a selection based on subcategorization of SM abnormal dogs (SM1 and SM2) [24]. This was also an important reason for us to use normal-versus-abnormal classification groups for further analysis of the relationship with qualitative assessments and quantitative measurements.

Of note is that the interobserver reliability for SM classification as normal versus abnormal for both the experienced versus novice observers and the experienced versus experienced observers was better than that previously reported in research on various dog breeds using low-field MRI studies [25]. This likely reflects the differences in image quality and the influence thereof on the assessment of the presence or absence of SM.

A key consideration in interpreting of the interobserver reliability in this study is that the novice observer received no particular training for their assessments in this study, other than being provided the information listed in the materials and methods section. As such, the interobserver reliability reported here is representative of the minimum interobserver reliability that can be obtained when comparing a novice versus an experienced observer. Some veterinary diagnostic imaging studies have reported intra- or interobserver reliability for various features relevant to the craniocervical junction of dogs that employed some sort of training before assessment of the study data [26,27], while many others do not mention if training was employed or not. Such training improves intra- or interobserver reliability, as differences between observations or observers will be ‘smoothed out’ before the actual assessments. Only 7.5% of human radiologic studies on agreement employed training for preparation of observers, based on a recent study analysis [28]. We elected to not include pre-assessment training for the novice observer.

### 4.2. MRI-Based Qualitative Parameters

While substantial interobserver agreement was found for the presence of CSF at the ventral aspect of the cervicomedullary junction, the agreement for the presence of the CSF signal between the cerebellum and the supraoccipitum and the presence of CSF at the dorsal aspect of the cervicomedullary junction was only slight. Intra-observer reliability was much better (substantial to almost perfect). As the interobserver agreement for two of the three parameters was only slight, there seems to be considerable subjectivity to these assessments. As loss of the CSF signal dorsal to the cervicomedullary junction would be an expected feature of CM, the low interobserver agreement between the novice and experienced observer for the CM classification can be partly explained by this subjective parameter.

### 4.3. Quantitative MRI- and CT-Based Measurements

For the MRI-based measurements, good interobserver reliability was found for only a few specific measurements, namely that of Area 1 (caudal cranial fossa), Area 2 (middle and rostral cranial fossa), and Angle 2 (the angle between the line from the caudal tip of the basioccipital bone to the cranial tip of the dens axis (red line, Figure 3) and the line from the caudal tip of the basioccipital bone to the midpoint of the caudal endplate of the axis (blue line, Figure 3). All other measurements showed a poor or moderate interobserver reliability. For the CT-based measurements, good to excellent interobserver reliability was found for all parameters assessed. Both MRI- and CT-based measurements showed good to excellent intra-observer reliability. Unsurprisingly though, intermodality agreement was only poor to moderate for MRI- versus CT-based measurements. As the measurements were all based on bony landmarks, it makes inherent sense that CT-based measurements would be more reliable.

Indeed, another study reported significant differences in the measurement of foramen magnum height in dogs based on CT versus MRI [29]. This finding sheds new light on previously reported findings in other breeds with CM/SM. A number of studies did not include multiple observers or did not include both CT and MRI imaging studies [4,5,8,10,26]. The interobserver and intermodality reliability of these different measurements or, indeed, the diagnosis of craniocervical junction disorders could be corroborated or shown by future studies.

### 4.4. Limitations

For all the MRI- and CT-based measurements and intermodality agreement analyses, it must be taken into account that not only the modality differed, but also, despite best attempts, the positioning of the patients. The dogs were positioned in sternal recumbency on the horizontal surface of the table with the head in a flexible coil (MRI) or a head rest (CT), both resulting in elevation of the head by about 2–3 cm above the table. Nevertheless, the extent of head and neck extension would have likely not been exactly the same for MRI and CT scans of each individual dog. As previous studies have shown that there is an effect of positioning on, for instance, vertebral alignment and position of the atlas, this would have affected intermodality variability in measurements [30]. Ideally, custom-made, individually molded positioning frames or cushions for each patient would prevent such intermodality positioning differences.

Another limitation is the fact that although multiple observers were included, the number of observers was limited. Future studies including more observers, with different levels of experience, may provide additional information on the variability of the results.

## 5. Conclusions

Inter- and intra-observer reliability for the classification of CM in Pomeranians is not perfect. CT-based morphometrical craniocervical measurements have higher inter- and intra-observer reliability than MRI-based measurements. These findings should be taken into account for clinical evaluation of imaging studies of Pomeranians with regard to CM/SM. Considering inter- and intra-observer reliability and agreement results is recommended for future studies on CM/SM in dogs.

## Figures and Tables

**Figure 1 animals-14-01854-f001:**
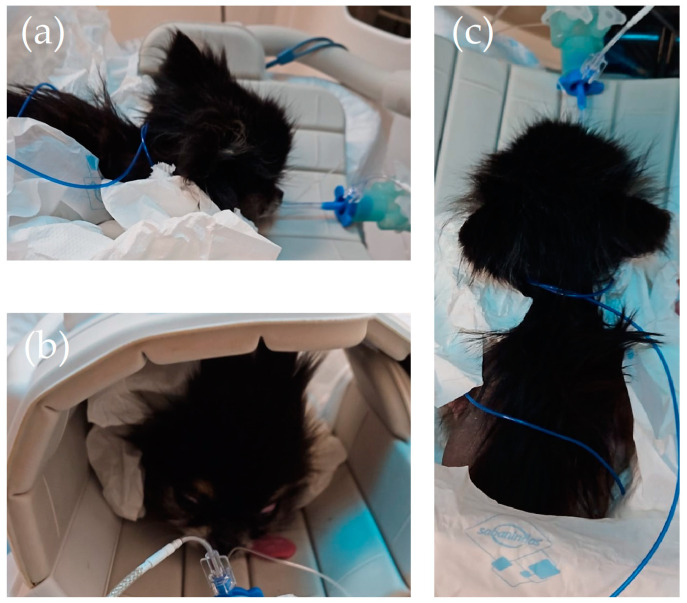
Positioning of the dogs for diagnostic imaging studies: (**a**) lateral view; (**b**) frontal view; (**c**) dorsal view. For CT scans, a head rest substituted for the flexible coil (in the pictures) used for MRI scans.

**Table 1 animals-14-01854-t001:** Characteristics of the study population.

Total Study Population	99 (100%)
**Sex**	
Female	47 (47%) 44 intact, 3 neutered
Male	52 (53%) 46 intact, 6 neutered
**Age ^1^**	2.9 years (1.9–3.7)
**Weight ^1^**	3.2 kg (2.6–3.9)

^1^ Median and interquartile range.

**Table 2 animals-14-01854-t002:** Kappa values and percentage agreement for interobserver agreement (experienced versus novice) on the classification of CM, the classification of SM, and MRI-based qualitative parameters.

Item	Kappa	95% CI ^1^	Percentage Agreement
CM classification	0.23	0.10–0.37	47%
CM normal/abnormal	0.31	0.12–0.51	70%
SM classification	0.50	0.35–0.64	69%
SM normal/abnormal	0.64	0.48–0.79	82%
Presence of CSF signal between the cerebellum and the supraoccipitum	0.11	−0.03–0.24	48%
Presence of CSF signal at the ventral aspect of the cervicomedullary junction	0.71	0.41–1.00	97%
Presence of CSF signal at the dorsal aspect of the cervicomedullary junction	0.20	0.00–0.39	62%
Syrinx location	0.61	0.49–0.73	75%

^1^ 95% confidence interval.

**Table 3 animals-14-01854-t003:** Intraclass correlation coefficients (ICC) for interobserver agreement on quantitative MRI- and CT-based measurements.

Item	ICC	95% CI ^1^
**MRI**		
Distance between the os tentorium cerebelli and the dorsum sellae	0.61	0.47–0.72
Length of the clivus	0.57	0.42–0.69
Height of the foramen magnum –‘foramen magnum line’	0.47	0.30–0.61
Distance between cranial tip of dorsal arch of the atlas and the foramen magnum line	0.59	0.44–0.704
Area 1 ^2^	0.80	0.71–0.86
Area 2 ^3^	0.86	0.793–0.901
Angle 1 ^4^	0.45	0.28–0.59
Angle 2 ^5^	0.75	0.65–0.82
**CT**		
Distance between the os tentorium cerebelli and the dorsum sellae	0.81	0.72–0.86
Length of the clivus	0.79	0.70–0.85
Height of the foramen magnum —‘foramen magnum line’	0.94	0.91–0.96
Distance between cranial tip of dorsal arch of the atlas and the foramen magnum line	0.95	0.919–0.963
Area 1 ^2^	0.76	0.67–0.84
Area 2 ^3^	0.92	0.85–0.95
Angle 1 ^4^	0.82	0.73–0.88
Angle 2 ^5^	0.76	0.66–0.83

^1^ 95% confidence interval, ^2^ Area 1 = area between the yellow line and the red line and osseous structures (caudal cranial fossa area, Figure 2), ^3^ Area 2 = area rostral to the yellow line (rostral and middle cranial fossa area, Figure 2), ^4^ Angle 1 = angle between the line from the caudal tip of the basioccipital bone to the cranial tip of the dens axis (red line) and the line from the ventral aspect of the supraoccipital bone to the cranial tip of the dens axis (green line) (Figure 3), ^5^ Angle 2 = angle between the line from the caudal tip of the basioccipital bone to the cranial tip of the dens axis (red line) and the line from the caudal tip of the basioccipital bone to the midpoint of the caudal endplate of the axis (blue line) (Figure 3).

**Table 4 animals-14-01854-t004:** Kappa values and percentage agreement for interobserver agreement (experienced versus experienced) on the classification of CM and SM as normal versus abnormal.

Item	Kappa	95% CI ^1^	Percentage Agreement
CM normal/abnormal	0.69	0.54–0.83	85%
SM normal/abnormal	0.82	0.71–0.93	91%

^1^ 95% confidence interval.

**Table 5 animals-14-01854-t005:** Kappa values and percentage agreement for intra-observer agreement (experienced) on the classification of CM, the classification of SM, and MRI-based qualitative parameters.

Item	Kappa	95% CI ^1^	Percentage Agreement
CM classification	0.79	0.67–0.91	89%
CM normal/abnormal	0.81	0.70- 0.93	91%
SM classification	0.95	0.90–1.00	97%
SM normal/abnormal	1.00	1.00–1.00	100%
Presence of CSF signal between the cerebellum and the supraoccipitum	0.76	0.62–0.90	90%
Presence of CSF signal at the ventral aspect of the cervicomedullary junction	0.79	0.51–1.00	98%
Presence of CSF signal at the dorsal aspect of the cervicomedullary junction	0.92	0.84–1.00	96%
Syrinx location	1.00	1.00 to 1.00	100%

^1^ 95% confidence interval.

**Table 6 animals-14-01854-t006:** Intraclass correlation coefficients (ICC) for intra-observer agreement on quantitative MRI- and CT-based measurements.

Item	ICC	95% CI ^1^
**MRI**		
Distance between the os tentorium cerebelli and the dorsum sellae	0.89	0.84–0.93
Length of the clivus	0.86	0.79–0.91
Height of the foramen magnum —‘foramen magnum line’	0.96	0.93–0.97
Distance between cranial tip of dorsal arch of the atlas and the foramen magnum line	0.99	0.98–0.99
Area 1 ^2^	0.89	0.83–0.92
Area 2 ^3^	0.93	0.90–0.96
Angle 1 ^4^	0.74	0.62–0.83
Angle 2 ^5^	0.99	0.98–0.99
**CT**		
Distance between the os tentorium cerebelli and the dorsum sellae	0.93	0.90–0.95
Length of the clivus	0.92	0.88–0.94
Height of the foramen magnum —‘foramen magnum line’	0.96	0.94–0.97
Distance between cranial tip of dorsal arch of the atlas and the foramen magnum line	0.98	0.98–0.99
Area 1 ^2^	0.98	0.96–0.98
Area 2 ^3^	0.97	0.96–0.98
Angle 1 ^4^	0.93	0.90–0.96
Angle 2 ^5^	0.95	0.94–0.97

^1^ 95% confidence interval, ^2^ Area 1 = area between the yellow line and the red line and osseous structures (caudal cranial fossa area, Figure 2), ^3^ Area 2 = area rostral to the yellow line (rostral and middle cranial fossa area, Figure 2), ^4^ Angle 1 = angle between the line from the caudal tip of the basioccipital bone to the cranial tip of the dens axis (red line) and the line from the ventral aspect of the supraoccipital bone to the cranial tip of the dens axis (green line) (Figure 3), ^5^ Angle 2 = angle between the line from the caudal tip of the basioccipital bone to the cranial tip of the dens axis (red line) and the line from the caudal tip of the basioccipital bone to the midpoint of the caudal endplate of the axis (blue line) (Figure 3).

**Table 7 animals-14-01854-t007:** Intraclass correlation coefficients (ICC) for intermodality agreement of quantitative MRI- and CT-based measurements.

Item	ICC	95% CI ^1^
Distance between the os tentorium cerebelli and the dorsum sellae	0.41	0.23–0.57
Length of the clivus	0.67	0.55–0.77
Height of the foramen magnum —‘foramen magnum line’	0.54	0.38–0.66
Distance between cranial tip of dorsal arch of the atlas and the foramen magnum line	0.22	−0.08–0.47
Area 1 ^2^	0.35	−0.10–0.68
Area 2 ^3^	0.67	0.01–0.87
Angle 1 ^4^	0.34	−0.07–0.61
Angle 2 ^5^	0.48	0.24–0.65

^1^ 95% confidence interval, ^2^ Area 1 = area between the yellow line and the red line and osseous structures (caudal cranial fossa area, Figure 2), ^3^ Area 2 = area rostral to the yellow line (rostral and middle cranial fossa area, Figure 2), ^4^ Angle 1 = angle between the line from the caudal tip of the basioccipital bone to the cranial tip of the dens axis (red line) and the line from the ventral aspect of the supraoccipital bone to the cranial tip of the dens axis (green line) (Figure 3), ^5^ Angle 2 = angle between the line from the caudal tip of the basioccipital bone to the cranial tip of the dens axis (red line) and the line from the caudal tip of the basioccipital bone to the midpoint of the caudal endplate of the axis (blue line) (Figure 3).

## Data Availability

The raw data supporting the conclusions of this article will be made available by the authors on request.

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
