# Peer review of "Craniocervical Morphometry in Pomeranians—Part I: Intra-Observer, Interobserver, and Intermodality (CT vs. MRI) Agreement"

_animals, 2024, doi:10.3390/ani14131854_

Round 1

Reviewer 1 Report

Comments and Suggestions for Authors

Dear Authors

I reviewed the manuscript entitled "Craniocervical morphometry in Pomeranians – Part I: Intra-observer, interobserver and intermodality (CT vs. MRI) agreement" . The manuscript describes the intra-observer, interobserver agreement and reliability of two modalities (CT vs MRI) of different measurements of the  morphology of skull and craniocervical region in normal Pomeranian dogs and Pomeranians with Chiari malformation and syringomyelia. The topic in interesting, since it describes how to perform the evaluation and the measurements in order to diagnose CM and SM. I have only minor comments, therefore I recommend minor revision

Specific comments

Materials and methods

line 92-93 Dogs were  positioned in sternal recumbency on the horizontal surface of the table with the head in a flexible coil (MRI) or a head rest (CT). Please describe more thoroughly how dogs were positioned, in order to reproduce the measurements (for example which kind of head rest, etc). Perhaps an image would be helpful.

line 103-104: For the CT scans, 0.6 and 0.8 mm slices were obtained with 130 kV and 80 mAs/slice and 130 kV and 220 mAs/slice and reconstructed in dorsal and sagittal  planes and 3D for further analysis. Please expand CT protocol. Rotation tube ? Pitch? Algorithm? 

Did the Authors try to evaluate SM also in CT images with HU measurements?

Were the MRI measurements performed only in T2W images?

Author Response

Dear reviewer,

Our sincere thanks for your time and effort spent to review this manuscript.

For line 92-93: We have included a new figure (photographs) that shows the positioning of the dogs.

For line 103-104: We have expanded the information on the CT protocol.

We did not evaluate SM on CT images with HU measurements (we decided not to as MRI is the gold standard and a previous study already compared MRI versus CT - https://pubmed.ncbi.nlm.nih.gov/32329949/).

All MRI sequences included in the methods were available for the observers and observers were allowed to perform measurements on the images side by side to facilitate the most accurate measurements (exact placement, of course, left to the observer in question). We have included an additional sentence in section 2.1. to clarify this.

Kind regards,

Authors

Reviewer 2 Report

Comments and Suggestions for Authors

The purpose of this article is to compare the statistical relationship of intraobserver, intraobserver and intramodality of MRI and CT findings of dogs in the context of Chiari malformation and syringomyelia. A total of 99 Pomeranian dogs were used for the study, and the examinations were evaluated by 3 radiologists with different experience. The results obtained can serve as a model for further research in this area. The article is divided into right standard paragraphs. In the introduction section, the authors briefly characterize the research problem undertaken and argue the validity of the topic.  In the materials and methods section, the authors correctly indicate the exclusion criteria for the study , the methodology for performing imaging and evaluating the results obtained, and describe the statistical analysis in detail. The figures presented are an important part of the described paragraph, however, they need to be more clearly labeled, as the lines in Figure 2 are poorly legible.  In the results section in the first sentence there is information about the exclusion of 13 animals , however, the reasons are not indicated. The rest of the results are presented in great detail, but in a clear and understandable manner. The discussion section is formulated interestingly and correctly. The conclusions , are rather poor and should be improved . The literature is up to date and correct. 

In conclusion, the article is interesting and provides a good base for researchers for further research, but it needs minor corrections before publication. 

Author Response

Dear reviewer,

Our sincere thanks for your time and effort spent to review this manuscript.
We address your comments below:

The purpose of this article is to compare the statistical relationship of intraobserver, intraobserver and intramodality of MRI and CT findings of dogs in the context of Chiari malformation and syringomyelia. A total of 99 Pomeranian dogs were used for the study, and the examinations were evaluated by 3 radiologists with different experience. The results obtained can serve as a model for further research in this area. The article is divided into right standard paragraphs. In the introduction section, the authors briefly characterize the research problem undertaken and argue the validity of the topic.  In the materials and methods section, the authors correctly indicate the exclusion criteria for the study , the methodology for performing imaging and evaluating the results obtained, and describe the statistical analysis in detail. The figures presented are an important part of the described paragraph, however, they need to be more clearly labeled, as the lines in Figure 2 are poorly legible. 

RESPONSE: We have addressed your comment concerning the lines in the figures. Thank you for pointing this out to us.

In the results section in the first sentence there is information about the exclusion of 13 animals , however, the reasons are not indicated.

RESPONSE: We included a specification for exclusion of these dogs.

The rest of the results are presented in great detail, but in a clear and understandable manner. The discussion section is formulated interestingly and correctly. The conclusions , are rather poor and should be improved . The literature is up to date and correct.

RESPONSE: We have rephrased the conclusions section (section 5).

In conclusion, the article is interesting and provides a good base for researchers for further research, but it needs minor corrections before publication.

RESPONSE: Thank you again for your time, effort, and constructive feedback!

Kind regards,

Authors